# Focused Ion Beam Processing for 3D Chiral Photonics Nanostructures

**DOI:** 10.3390/mi12010006

**Published:** 2020-12-23

**Authors:** Mariachiara Manoccio, Marco Esposito, Adriana Passaseo, Massimo Cuscunà, Vittorianna Tasco

**Affiliations:** 1Department of Mathematics and Physics Ennio De Giorgi, University of Salento, Via Arnesano, 73100 Lecce, Italy; 2CNR NANOTEC Institute of Nanotechnology, Via Monteroni, 73100 Lecce, Italy; adriana.passaseo@nanotec.cnr.it (A.P.); massimo.cuscuna@nanotec.cnr.it (M.C.); vittorianna.tasco@nanotec.cnr.it (V.T.)

**Keywords:** focused ion beam milling, focused ion beam induced deposition, 3D nanostructuring, chirality, chiral photonics, circular polarization, chiroptical effects

## Abstract

The focused ion beam (FIB) is a powerful piece of technology which has enabled scientific and technological advances in the realization and study of micro- and nano-systems in many research areas, such as nanotechnology, material science, and the microelectronic industry. Recently, its applications have been extended to the photonics field, owing to the possibility of developing systems with complex shapes, including 3D chiral shapes. Indeed, micro-/nano-structured elements with precise geometrical features at the nanoscale can be realized by FIB processing, with sizes that can be tailored in order to tune optical responses over a broad spectral region. In this review, we give an overview of recent efforts in this field which have involved FIB processing as a nanofabrication tool for photonics applications. In particular, we focus on FIB-induced deposition and FIB milling, employed to build 3D nanostructures and metasurfaces exhibiting intrinsic chirality. We describe the fabrication strategies present in the literature and the chiro-optical behavior of the developed structures. The achieved results pave the way for the creation of novel and advanced nanophotonic devices for many fields of application, ranging from polarization control to integration in photonic circuits to subwavelength imaging.

## 1. Introduction

The development of systems of increasing complexity, enabled by the advanced nanofabrication technologies which are available today, is increasing the potential of nanophotonics [1]. Metamaterials and photonic crystals with accurate and tunable geometries at the nanoscale level can now be manufactured, broadening the scope of possibilities from traditional planar geometries to complex systems, and making it possible to explore new optical properties and to modulate light–matter interactions in specific spectral regions [2,3,4]. 

One example is represented by chiral optical systems. Chirality is a geometric feature which is characterized by a lack of specular symmetry. It is present in many forms in nature, such as molecules, proteins, our hands, and even galaxies [5]. This geometric feature is of great importance in the study of chemistry and biology because two enantiomers (i.e., two specular images of each other) may have different physiological responses, even though they do not seem to differ in terms of their chemical and (almost all) physical properties. As an example, one enantiomer can provoke toxic pharmacological effects while another is innocuous [6]. However, the only way to distinguish between the two turns out to be the interaction of the molecules with circularly polarized light (CPL). Thus, study of the optical responses of these systems is very important, because it allows researchers to distinguish between two apparently identical enantiomeric forms. However, chiro-optical effects are very faint in molecules because of their small dipole moments, and they are generally observed only in the ultraviolet (UV) region. Artificial chiral photonic structures with nanoscale dimensions, mimicking natural chiral molecules, can be engineered in such a way that these effects are boosted, making them observable also in the VIS and IR range [7]. These structures have been applied in a broad range of investigation fields, from the sensing of biomolecules to miniaturized and integrated devices [8], as we will see in the following paragraphs. Chiral metamaterials [9,10] and chiral plasmonic nanostructures [11,12] exhibit extraordinary optical properties [13,14] which are strongly enhanced when they are designed in three dimensions, where geometrical chirality is more pronounced [15]. However, the fabrication of 3D nanostructures with intrinsic chirality is very challenging, because complex shapes such as spirals [7] must be realized with a high level of accuracy and spatial resolution to ensure reliability and reproducibility in optical experiments [16]. New strategies to fabricate complex 3D chiral objects at the micro- and nano-scales are nowadays in great demand. Focused ion/electron beam technology has demonstrated the ability to manipulate materials at the nanometer scale by directly patterning the substrate to create nanostructures with a high degree of precision [17] and without limitations in terms of structural complexity, making it possible to include 3D and chiral features. Compared to other nanofabrication techniques, focused ion/electron beam technology stands out for many reasons, such as its enormous flexibility in 3D design, but also because it requires only a single technological step, is compatible with many materials, and requires little preparation time [18]. Focused ion beam (FIB) technology found its first applications in the semiconductor industry [19], in the repair of defects in lithography photomasks [20,21], in circuit modifications [22], in transmission electron microscope (TEM) sample preparation [23,24], and in failure analyses [25]. Subsequently, FIB processing has been used for the fabrication of tools for atomic force microscopy (AFM) [26] and scanning optical near-field (SNOM) microscopy [27]. Nowadays, its fields of application have been extended to include various research areas requiring nanometer-scale imaging, lithography, material removal, and deposition. The recent progress of this technology has allowed entry into the nanophotonic field of research [28], which was not accessible before, mainly because of limited control of the process yield at the nanoscale, along with fundamental issues related to material composition.

In this review, we will explore recent progress in the realization of 3D chiral structures using FIB technology for optics and photonics applications, considering both the top-down etching process of ion milling and the bottom-up growth process of ion beam-induced deposition. We will present the advantages and disadvantages of both techniques, e.g., the ability to mill almost any type of material, and the possibilities offered by induced deposition, which has yielded nanostructures presenting significant chiroptical effects, regardless of the chemical composition of the deposited material. It is worth noting that we will not consider the development of focused electron beam technology for nanophotonics, as this topic will be treated elsewhere in the present Special Issue.

This review is organized as follows. In Section 2, we will present a brief overview of the basic operating principles of FIB milling and induced deposition (FIBID) employed in nanofabrication. We will explore the possibilities associated with the use of different types of ion sources and the advantages of highly-localized fabrication. In Section 3, we will discuss material composition issues in photonics nanostructures realized by FIB technology, and will present some purification approaches proposed in the literature. In Section 4, we will show ion milling and FIB-induced deposition strategies for the realization of 3D chiral photonic nanostructures, discussing the observed chiroptical effects. In Section 5, we will conclude this paper with a brief comparison with other, competing techniques, as well as discussing the possible future evolution of this technique.

## 2. Focused Ion Beam Processing

In recent years, FIB processing has gained much interest for the creation of prototypes of three-dimensional nanostructures for applications in photonics, thanks to its very high spatial resolution and flexibility regarding structural design. With FIB processing, the fabrication of complex nanostructure, can occur through material removal caused by the ion milling capability, or through local deposition induced by the interaction between the ion beam and a gaseous precursor.

Remarkably, this technology can work with both material families of interest for nanophotonics, i.e., conductive and lossy metals, and insulating and low losses dielectrics. FIB processing is usually carried out in a vacuum chamber, equipped with both electron and ion optical columns even though, in this review, we are focusing on the ion beam-processing. The high energy beam is emitted from an ion source, accelerated at energies ranging from 5–50 KeV, and focused on the sample surface by means of a series of electrostatic lenses. By adjusting the current passing through the lens, the beam can be finely focused, achieving a spot size as narrow as 10 nm [29].

When the ion beam interacts with the sample surface, many phenomena occur [30]:Emission of secondary electrons, usually employed for sample imaging;sputtering of the substrate atoms;deposition in the presence of gaseous precursors in the chamber; andre-deposition of some sputtered atoms and ion implantation from the beam, leading to amorphous/damaged/rough surfaces.

The design and manufacturing of the nanostructures with focused ion beams is ruled by several parameters which define the resolution, the pattern dimension (pixels), and the quality of the structures. These parameters are: the ion species, the ion dose, the incident angle, the beam energy, and the accelerating voltage. The nanofabrication is also driven by pattern parameters, such as the dwell time (i.e., the time during which the ion beam stays in one position), and the step size (i.e., the distance between two consecutive beam positions).

Moreover, the patterning of complex geometries at micro- and nano-scale takes advantage of computer-aided design and manufacturing systems, particularly helpful in improving dimensional accuracy and reproducibility during fabrication [31].

The focused ion beam (either with heavy ions like Ga^+^ [32], or light ions like He^+^ [33]) can be also employed as a scanning ion probe for lithographic patterning in resists, with position and timing controlled by a pattern generator. The ion beam lithography reaches higher resolution as compared to EBL, even if with the same spot size, thanks to the absence of backscattering effects together with a weaker forward scattering and a smaller lateral diffusion of secondary electrons [34]. Moreover, the ions, heavier than electrons, can penetrate the sample with higher energy, allowing a faster exposure of the resists and a faster processing [34,35] of photonics nanostructures.

In the next sections, we will explore the features of FIB processing and the manifold strategies implemented to construct complex and 3D photonic structures.

### 2.1. Basics of FIB Milling

A scheme of the FIB milling process is reported in Figure 1a. During the process, which is highly destructive for the sample, the ion beam locally scans the surface digging the targeted area. The milling of the surface can be integrated with gas-assisted etching if gaseous organic precursors are simultaneously introduced in the vacuum chamber through a gas injection system (GIS) [36].

In the milling process, the sputtering rate is ruled by the energy of the beam, which hits and locally removes the substrate atoms. During the sputtering removal, the ions are implanted into the sample.

The most used source is the gallium liquid metal ion source (LMIS) [37]. Alternative and less common LMIS based on other metals, such as B, Be, Si, Sn, Au, Fe, Ni, Cr [38], and alloys like PdAs, PdAsB, AuSi, and AuSiBe, have been also developed and are currently studied [39,40].

The liquid metal lies in contact with a tungsten tip. When applying a high voltage, the heated metal wets the tungsten tip, generating a strong electric field on the tip, which leads to the ionization and the emission of the gallium beam. Ga^+^ LMIS is considered the most advantageous solution because of its low melting point (303 K) and low vapor pressure [38]. A focused Ga^+^ beam can hit the sample with current ranging from 1 pA to 10 nA, with minimum spot size smaller than tens of nanometer [41,42]. However, the sputtered gallium ions cause significant material redeposition and ion implantation on the substrate. To avoid surface damaging and ion implantations, the gas field ionization sources (GFIS) can be employed [43]. These sources are based on the ionizations of helium or neon, which have smaller atomic sizes than gallium. GFIS have been optimized for fabricating nanostructures with very small geometric features, such as nanopores and nanoribbons [44,45].

A He^+^ FIB system provides very small milling currents (0.1 nA) and an ultra-narrow beam spot of less than 0.5 nm [46] with milling resolution of 3.5 nm [47]. The direct He^+^ FIB milling of an Au film was able to produce high-quality nanostructures, such as plasmonic systems consisting of seven close-packed holes, with a 100 nm diameter and very sharp edges, separated from each other by only 15 nm, [48] as shown in Figure 1b. Moreover, in this case, a limited effect of material redeposition from the substrate has been observed.

Other types of sources, largely employed in plasma-based FIB microscopes, are noble gases of heavy ion species, such as Xe or Ar. They can deliver high current beams up to microampere and, because of their larger size, they can be employed for the fast milling of large volumes (up to hundreds of cubic micron) [49].

Since the ion beam is highly localized, it allows for the integration of plasmonic nanostructures on small structures, such as the tips of optical fibers, inducing light-manipulation capabilities as plasmon waveguides [50]. Recently, this concept has been extended also to metasurfaces integrated on optical fibers, for the promising perspectives of optical metasurface sensors on fiber [51].

### 2.2. Basics of FIBID Growth Dynamics

The first approaches of controlled focused ion/electron beam induced deposition (FIBID/FEBID) growth were investigated in [17,36,52,53], paving the way to the micro/nanofabrication of precise and complex-shaped 3D structures. Today, the availability of different gas precursors [36] to be used in conjunction with the beam is enabling a wide gamut of optical, magnetic, superconducting, and mechanical properties [52,54,55,56].

FIBID procedure works on the principle of local chemical vapor deposition (CVD) [41]. The ion beam breaks the gas precursor molecules coming from the GIS, and leaves a deposit on the substrate, acting as a nucleation site for the nanostructure growth, as schematically illustrated in Figure 1c. Once the design and the proper growth parameters are defined, the precursor decomposition occurs in the targeted area irradiated by the beam, following the beam pattern. The evolution in the third dimension occurs when the beam reaches the edge of the fabricating structure. The secondary electrons emitted during the proceed enhance the lateral growth on the sample [30,57].

There are many physical mechanisms generated by the interplay with the beam and the substrate which influence the growth in the third dimension. These include secondary electrons emission, scattered particles, and charge effects. In the case of the consecutive growth of several elements, proximity effects are generated due to the bending of the electrostatic force between two neighboring structures. It has been shown that during the growth of two consecutive metallic pillars, the second pillar becomes taller than the first one, while the first pillar becomes broader and slightly folds toward the second one. The studies on proximity effects of Pt-based pillars grown on a Si_3_N_4_/Si wafer [58] have shown that the scattered ions and the emitted secondary electrons and atoms produced during the second pillar growth induce additional deposits for the first pillar, which, in turn, broadens its diameter. Consequently, the amount of the deposited material induced by proximity effects varies with the separation gap, as shown in Figure 1d. However, for more complex nanostructures like 3D chiral nano helices [54], proximity effects also occur during the growth of the single structure, because of a larger interaction volume of the spiral with the radially scattered particles (schematic illustration of Figure 1e), as compared to the nanopillar case. Here, the proximity effects promote the precursor decomposition close to the substrate, leading to a gradual loop height reduction along the nanostructure, as we will better describe in Section 4.

Moreover, the number of scattered particles is also related to the substrate material: The more conductive the substrate is, the more effective the charge effects are, increasing the growth rate and influencing the final sizes. Thus, the employment of a conductive substrate can limit the charge effects for a better growth control.

Moreover, the values of step size and dwell time must be also adjusted: The former to control the density of the nucleation sites, and the latter to define the amount of deposited material during each deposition spot. It is worth noting also the role played by the local pressure of the gas precursor and the distance of the GIS from the substrate [59]; both influence the amount of the material that will be deposited.

Focused electron beam induced deposition is a technique very close to FIBID, but employs a high energy electron beam generated through a scanning electron microscope.

The growth mechanisms and the related properties are widely explored in literature [36,52,60,61]. We just remind that the most important difference between FIBID/FEBID growth of nanostructures consists of the achievable size and composition. In particular, FEBID allows for further reduction of the dimension of the fabricated nanostructures because of the smaller electron size, as compared to ions. Moreover, while the ion beam causes ion implantation that affects the final composition of the fabricated structures, as we will show in the next sections, this problem is absent in FEBID structures.

### 2.3. FIB under Cryogenic Conditions

A recent upgrade concerns FIB processing under cryogenic conditions, which cause the condensation of the precursor material on the substrate. Cryo-FIBID processes were realized and studied using tungsten [62] and platinum-based [63] precursors. The cryogenic temperatures can help to retain the microstructures of sensitive materials when performing milling treatments for electron microscopy [64]. For FIB induced deposition, the low process temperatures cause an ultrafast growth and, at the same time, reduce the proximity effects and the ion implantations [65] improving material composition [63].

## 3. Material Features of Nanostructure FIB Processing

### Composition Assessment

A fundamental feature that so far hindered the widespread use of FIB processing in nanophotonics is the complex material composition of the fabricated nanostructures. FIB processing results in a low purity level [66], caused by the interaction between the ion beam with the substrate, because of the ion implantations and sputtered atoms redeposition. The low material purity is correlated with the incomplete dissociation of the gas precursor molecules, which leaves residual percentages of carbon and oxygen in the deposit. In addition, it is worth considering the contamination coming from the volatile residual species like CO and CO_2_ present in the vacuum chamber, which contribute to the high carbon percentage, always recorded by the composition studies of the patterns/structures. The process parameters can also have a role in controlling gallium implantation, sputtered atom redeposition, and carbon percentage [67]. The presence of impurities might constitute a limitation on the functionality of the structures with respect to plasmonic applications, which require pure metallic surfaces. At the same time, the implantation of sputtered atoms and beam ions can introduce absorption losses in the optical response of otherwise transparent materials. Moreover, robust numerical and analytical modeling to understand and predict the structure optical behavior should rely on handbook-level or well-defined dispersions for the employed materials. Ideally, local optical investigation should be performed directly on FIB nanostructures to access the actual material dispersion, but this represents a challenging experimental issue, especially for chiral objects.

The compositional analysis of nanostructures manufactured by FIB processing (both milling and deposition) is usually performed by Energy Dispersive X-Ray Spectroscopy (EDS), transmission and scanning transmission electron microscopy (TEM and STEM respectively), Raman spectroscopy, Fourier Transform Infrared (FTIR) Spectroscopy, Auger electron spectroscopy (AES), and FIB cross section preparation [36].

During the milling process, the impurities introduced by the Ga+ implantations and re-deposition of the sputtered atoms can alter the fabricated nanostructure damaging the substrate [30]. Gallium implantation is detrimental in dielectric structures, increasing the absorption losses. This is the case of the 3D chiral photonic silicon platforms fabricated on sapphire (SOS) [68], shown in Figure 2a. Here, the STEM images, performed after the fabrication, demonstrated the presence of gallium implanted on a layer of damaged silicon (d-Si), and amorphous silicon inclusions (c-Si) (Figure 2b). The platforms have shown optical transmission lower than 20%. In order to remove impurities after the FIB milling process, a thermal oxidation treatment, followed by annealing, was carried out. The STEM image taken after the treatment (Figure 2c) demonstrated that the damaged silicon layer was replaced by silicon oxide. This cleaning procedure allowed for the improvement of the transmission intensity up to 70%. Moreover, the method proved advantageous for the fabrication of pure 3D chiral dielectric metasurfaces, exhibiting circular polarization discrimination in transmission in the visible range.

FIBID technology relies on a huge variety of available precursors, and consequently, materials that can be deposited. Organometallic precursors containing Pt, Au, Cu, and W have been originally developed for creating electric contacts and for mask repair. Nowadays, precursors of the noble metals, like gold or platinum, can be employed for plasmonic applications [69], while W precursors can serve for studies on superconductivity in FIBID nanostructures [56]. Precursors of magnetic materials (like cobalt) are of great interest for the fabrication of nanosensors, nanodevices, and for fundamental studies on nanomagnetism [55]. Many FIBID precursors have been also developed for carbon [70], which can be deposited in different forms, depending on growth conditions (39) (amorphous carbon, graphite, diamond, and diamond like carbon (DLC)) [71]. SiO_2_ precursors for FIBID, like tetraethyl orthosilicate (SiC_8_H_20_O_4_), have been used to deposit oxide films which displayed superior insulating properties and with low contamination [72].

Detailed compositional studies have been carried out on microstructures and thin films from organometallic precursors of platinum, copper, and gold [73,74,75]. In these cases, it was found that the deposits consist of metal nanocrystals, uniformly embedded in a matrix of amorphous carbon, exhibiting mechanical stability and chemical protection. A HRTEM image of FIBID Pt film deposited on a Cu grid is shown in Figure 2d, where the observed dark/bright contrast confirms the distribution of Pt nanoparticles immersed in the amorphous carbon matrix. Crystalline atomic planes and their related diffraction spots have been resolved, demonstrating the crystalline nature of metal grains [76]. These results demonstrated to be similar for Pt-based nanowires studied in other works [66,77].

Another representative case of the complex material architecture achieved by FIBID is when the Ga+ beam is used in conjunction with the precursor of a low molecular weight compound. Such a behavior has been firstly observed for pillar fabricated with phenanthrene (C_14_H_10_) and other carbon gaseous precursors [67,70,78]. Because of the gallium scattering length of 20–30 nm into the solid carbon [79], Ga ions are implanted in the nanostructure core, while the carbon is mainly confined in the outer part of the nanostructure forming a C-based shell. Figure 2e shows the TEM image of a carbon pillar, where the dark central part corresponds to the Ga core located at the center of the pillar, and the external part represents the amorphous carbon shell [80].

As an example of the compositional complexity that can be attained by FIBID, Figure 3 shows how the same nano-helix shape, realized by using three different precursors, can have completely different material architectures [81,82]. In particular, (methylcyclopentadyenil)platinum(IV) has been used for Pt, phenantrene (C_10_H_10_) for C, and TEOS (Si(OC_2_H_5_)_4_) for SiO_2_. The high magnification of High Angle Annular Dark-field (HAADF) STEM image (Figure 3b) of the Pt-based nano-helix (Figure 3a) revealed an amorphous carbon matrix in which platinum metallic incursions (with averaged size 5 nm) are uniformly embedded. The distribution of elements in volume percentage throughout the nano-helix section is: 50% platinum, 45% carbon, and 5% of implanted gallium (Figure 3b,c). It is worth noting that the Pt nanograins are placed close to each other, leading to an overall metallic behavior, as discussed below.

The HAADF-STEM image of a nano-helix (Figure 3d), grown by using phenanthrene as a carbon precursor, exhibits a core-shell profile with an inner core composed by gallium ions and an amorphous carbon shell, similarly to what was discussed before for carbon pillars and found in [74]. In particular, in the helix case, Ga nanoparticles concentrate inside a narrow core, with a volume percentage of 65%, whereas the thick outer shell is composed by 95% C and 5% Ga (Figure 3e,f). The SiO_2_ nano-helix (Figure 3g) also exhibited a Ga-rich core (68% Ga/28% SiO_2_/4% C) and a SiO_2_ -based shell (8% Ga/88% SiO_2_/4% C). Even though the same growth parameters and structural features have been used, the Ga-core in the latter case is thicker than the former (Figure 3h,i). This happens because the Ga implantation in SiO_2_ starts at the surface, given the large interaction between gallium ions and Si nuclei [83].

Starting from the structural composition observed by TEM/EDX analysis, a customized numerical model can be developed to retrieve the artificial material dispersions. The procedure was performed for the Pt-based structures (Figure 3j) and for both the core and shell of C and SiO_2_ (Figure 3j,k, respectively) at the visible frequencies. The simulated effective refractive indexes (n) and absorption coefficients (k) underline the metallic behavior of the platinum wire, and of the gallium core, for both C and SiO_2_ nano-helices in the visible range. On the other hand, the two shells exhibit a dielectric behavior with a lower absorption for the SiO_2_-based shell.

The metal selection is fundamental for the generation of high-quality localized surface plasmon resonances (LSPRs) in nanostructures. Therefore, the material complexity of chiral FIBID/FEBID nanostructures can reduce their chiro-optical performances. In FEBID nanostructures, where gallium implantation does not occur, purification approaches of the deposited materials have been proposed just to remove the residual carbon content, with satisfactory results. These methods are: In situ substrate heating during the deposition process, post-treatment annealing in oxidizing atmospheres [84], electron irradiation or laser treatment of the structures [85], oxygen plasma [86], or ozone [87]. Another strategy, applied to both FIBID and FEBID nanostructures, consists of a post-process metal coating by means of thermal evaporation or sputter coating [59,88,89,90]. In this way, a metallic shell, thicker than the plasmonic skin depth, improves the plasmonic response from FIBID-based metal nanostructures [90].

Very few studies have been performed on FIBID structures, to our knowledge. Recently, a purification approach employing an oxygen flux during deposition was applied to planar platinum-based pads with 200 nm thickness [91]. Oxygen is suitable to form volatile species like CO and CO_2_ and, thus, to reduce the carbon amount caused by the growth dynamics. Here, two different methods, both in situ and at room temperature, were applied: One is a post-deposition irradiation of a pad under O_2_ flux; the second one consists of the simultaneous injection of oxygen and platinum precursors during the deposition. In both cases, the deposition occurred under the same growth conditions. Both experiments demonstrated a reduction of C/Pt ratio with a purity level close to bulk Pt. However, in the post deposition treatment, despite the efficient carbon removal, an increase of oxygen amount has been detected in the final nanostructure composition.

However, as we will show in the next section, the impurities are not necessarily harmful. As an example, it is possible to benefit from the effects of stress and strain caused by ion implantations on substrates during FIB milling to promote the fabrication of three-dimensional structures with broken symmetry [92]. Moreover, new frontiers in optics currently aim to research new plasmonic materials beyond noble metals [93]. For example, large chiroptical effects have been demonstrated as arising from the gallium plasmonic core in the Ga/SiO_2_ core/shell nanohelices discussed above [82].

## 4. 3D Chiral Nanostructures Realized by FIB Processing

Planar or two-dimensional nanostructures can demonstrate chiro-optical effects because of extrinsic chirality arising from specific geometrical conditions of the external illumination. However, the employment of 3D (or quasi-3D) structures is fundamental to get intrinsically chiral shapes and to maximize the chiral properties as compared to the planar counterpart.

Original strategies based on FIB processing have been implemented to create new chiral geometries with evolution along the third dimension, in order to maximize the chiro-optical behavior.

As we have already mentioned, FIB processing consists of two important tools: Milling and induced deposition. As we will see in the next paragraphs, both can be used to realize complex and chiral forms. On one hand, very recently, various top-down milling strategies have been developed to create large-area, 3D, and quasi-3D metasurfaces in metallic films with broken symmetry and intrinsic chirality. On the other hand, the induced deposition, as a bottom-up technique, allows the creation of three-dimensional nanostructured elements with the most varied shapes, in particular the helix shape, representing the ideal 3D chiral geometry. The most important features of FIB processing for single and periodic arrays of 3D chiral photonic nanostructures are: (i) The direct writing in a single step to speed up the fabrication procedure; (ii) the high resolution suitable to grow and mill element with size <100 nm; (iii) the large design freedom at the nanoscale and high flexibility in tuning the geometrical features to tailor the optical response in a broadband spectral range.

However, FIBID is still affected by limitations with respect to large-area fabrication of reproducible structures, and to writing speed. As a result, the fabrication areas practically achievable by FIBID are limited to tens of microns [69]. Instead, FIB milling is suitable for reproducing patterns which are even hundreds of microns large, with great repeatability and in a few hours [94].

The chiroptical effects of interest for chiral nanomaterials originate from the interaction of these objects with circularly polarized light (CPL), and are the circular dichroism (CD) and optical rotation dispersion (ORD) [95]. Circular dichroism is defined as the differential absorption between left and right-handed components of the CPL (LCP and RCP, respectively). In many cases reported in literature, a general CPL discrimination is reported with respect to measured transmitted or reflected light. Optical activity (or optical rotation) represents the rotation of linearly polarized light traveling through a chiral medium. In this section we will explore the efforts conducted to develop three-dimensional chiral nanostructures by means of FIB processing, and the related chiroptical effects in the visible and IR range.

### 4.1. 3D Chiral Nanostructures Realized by FIB Milling

#### 4.1.1. One-Step Tilted-Angle Focused Ion Beam Milling

The combination of FIB milling with engineered path and beam properties has demonstrated the fabrication of nanocomponents with large and spectrally tunable chiro-optical effects. For example, the induction of symmetry breaking on the nanostructures leads to an anisotropic response when interacting with left- and right-handed circularly polarized light. A one-step tilted-angle focused ion beam represents a good candidate to induce symmetry breaking [92]. Here, the sample stage is tilted with respect to the ion beam at slanted angle ϕ, as schematically illustrated in Figure 4a. Under this condition, plasmonic slanted split ring apertures (SSRA) with arbitrary shapes are simply created in a gold film of 180 nm. The SEM top views of two enantiomeric forms (A and B) of the SSRA nanoapertures arrays with period 400 nm are shown in Figure 4b, while Figure 4c shows the asymmetrical optical response to the two components of the circularly polarized light (LCP and RCP, respectively) in transmission. Strong circular dichroism in transmission (CDT) over 78% in the near-infrared wavelength range (Figure 4c) is observed, which can be easily tuned in a broad spectral with the in-plane rotation of the nanoapertures. Such a large CDT arises from the coupling between the circularly polarized light and the segments of the nanoaperture, acting as a waveguide under the incidence of left- and right-handed polarizations.

Moreover, the flexibility of this technique allows us to extend this concept to other designs like slanted L-shaped aperture and slanted rectangular aperture, for applications in optical information processing, chiral imaging, and sensing.

#### 4.1.2. FIB Milling and Gray Scale Bitmap

When using grayscale bitmap files as sources for the beam path, a large variety of geometries and milling profiles can be obtained.

For example, ramp-shaped gold split ring structures with a gradient height were realized uploading a gradient gray scale bitmap file to the FIB system. In this way, the planar and symmetric split ring geometry can be transformed in a structure with gradually varying height and broken symmetry [96]. The two enantiomeric forms of gold on glass slide depend on the increasing direction of the gradient depth (either clockwise or counterclockwise) and are shown in the inset of Figure 4d. The fabrication of both positive (removing the background and leaving the structure) and negative (removing the structure from the background) variants of the structure is possible. The gradient height and geometrical parameters are easy to modify and can create photonics devices like circular polarizers. Two-dimensional periodic arrays of such 3D ramp-shaped gold nanostructures, under the excitation of left and right-handed circularly polarized light, displayed large dissymmetry in reflection at the visible wavelengths, with values of circular dichroism in the VIS regime up to 64%, while transmission is close to zero, as shown in Figure 4d. The origin of the strong CD is not only the chirality induced by the mirror symmetry breaking, but also the additional anisotropy introduced by the irregularities of the nanofabrication process. In fact, the gradual height variation along the pattern from the deepest point to the highest is not very smooth (Figure 4e), and the edge profile is affected by unavoidable imprecisions caused by the FIB gradient milling.

Another example of 3D chiral structures realized with this technique are the V-shaped nanoapertures [97] on optically thick gold film (180 nm), obtained by truncating selectively only the right or the left half of the V-shaped nanoapertures. The flexibility of FIB technology allows us to apply different ion doses at the two halves of the V-shaped nanoaperture to produce different milling depths in the two sides. In this way, metasurfaces with broken symmetry can be realized. In Figure 4h, the scheme of the two stepped V-shaped nanoapertures with the two enantiomeric forms is shown, with the top and side views of the SEM images.

Optical metasurfaces have been studied for miniaturization and integration of devices capable of generating chiral images by switching on or off the pixel (depending on the polarization handedness) to display only binary images, with black and white colors with 1 bit pixel depth [98]. The employment of stepped V-shaped nanoapertures demonstrated improved spatial resolution, quality of the imaging, and high data density thanks to the ability to control chiral images with 8-bit pixel depth in grayscale colors.

Two enantiomeric arrays of the stepped V-shaped nanoapertures were designed to display the grayscale portrait images of Einstein and Marie Curie (Form A and B, respectively, as showed in Figure 4f), and then they are merged with tailored orientation angles. Each orientation angle of the stepped V shape nanoapertures was set to correspond to an image pixel (Figure 4g). Each unit cell interacts with the circularly polarized light, and each output polarization controls the shades of gray for generating grayscale images [99,100].

Moreover, these nanostructures were spatially arranged with the orientation angle determining the local polarization direction. Then, the desired intensity profile is converted through an analyzer following the Malus law. The stepped V-shaped nanoaperture can be considered as a two-channel optical spin filter cascaded by a photon extractor to switch the left or right channel. When interacting with CPL, the incident right-handedness is selectively focused into the channel with the opposite handedness, and then transmitted through the gold film, while the other polarization is back reflected.

The concept of one-step gray scale tailored by FIB milling parameters has been also used with 3D Janus plasmonic nano-apertures [94] shaped as helical nanostructures. The schemes of the two enantiomeric forms (A and B, respectively) are depicted in Figure 4h, along with the related top view and side view SEM images. These helical-like nanostructures have been developed on optically thick gold with gradually increasing groove depth, from 0 (not etched) to the total gold film thickness (totally etched surface). The ion beam dose gradually increases for deeper milling, as schematically shown in Figure 4i. Instead, the graph in Figure 4j shows the milling depth per unit of ion dose, which is gradually reduced with the increase of the groove depth, because of redeposition effect in the nanoapertures during the gradual dig [101]. However, this effect can be corrected by adjusting the beam parameters, like focus and astigmatism. The principle is based on encoding the Janus metasurfaces in a way that a binary image and a grayscale image can be separately displayed in the forward and backward directions under circular and linear polarization, respectively. The study of these surfaces displays a binary quick response (QR) code image in the forward direction under circularly polarized incidence of one handedness of the CPL, while showing another grayscale image in the backward direction under linearly polarized incident light (Figure 4h). As with the previous case, the rotation angle of the Janus nanoapertures corresponds to a grayscale pixel under linear polarized light, and then the intensity profile is converted with an analyzer according to Malus’ law. The optical properties of these metasurfaces suggest a possible employment for direction-controlled and polarization-encrypted data storage, data encryption and decryption, and optical information processing.

#### 4.1.3. Focused-Ion-Beam Stress-Induced Deformation Effects, Folding and Bending

A recently developed employment of FIB milling for 3D nanofabrication exploits mechanical effects caused by the collision of the ion beam on the substrate surface, such as bending, folding, or stress-induced deformations. By a proper control of the ion dose and the irradiated area, periodic complex 3D micro/nanostructures, even with chiral features [102,103], can be manufactured. The concept is based on transforming an unfolded 2D pattern into a 3D micro/nano system, and it is suitable for various metals and dielectric thin films. The FIB irradiation on a thin film can locally induce stress or strain fields, thus folding, bending, and/or lifting the thin film. These mechanical effects can be applied to obtain the desired shape, as a function of the fabrication parameters, including accelerating voltage, beam current, and ion irradiation time. Moreover, by controlling the irradiation dose, the damaging of the material can be controlled. FIB stress induced deformation (FIB-SID), to create arrays of 3D meta-atoms on a suspended gold thin film, were optically and mechanically studied in [102]. The reflectance spectra of the 3D meta-atoms have shown polarization dependence, when interacting with linearly polarized light, with spectral peaks located within the range of mid-wave IR and long-wave IR. This concept can be applied to more complex structures, like 3D chiral nano-helices. In [103], 3D aluminum helical optical antennas with smooth surfaces and with various geometrical parameters (radii ranging from 600 nm to 2.1 μm) were demonstrated thanks to the accurate bidirectional folding (−70°–+90°) provided by the technique. The fabrication process is shown in Figure 5a, where the tip of a 2D aluminum film with 100 nm thickness, suspended on a Si substrate, has been exposed to ion beam irradiation. This causes the folding of the Al stripe at an angle depending on magnitude of the incident ion dose. The process is repeated with different values of irradiation interval and irradiation dose. Each value causes the bending of the cantilever at a certain angle. When repeating this procedure several times and for different angles, the bending is performed several times. In this way, the 2D strip is gradually curled, forming the helix, whose SEM images are shown in Figure 5b. The observed optical rotation (graph in Figure 5c) measured for two helices with radius *R* = 1.2 μm suggested that the polarization direction could be rotated by a maximum value of 8° at 800 nm using these systems. Strong optical resonances have not been observed because of the large material losses at the selected optical region.

Very recently, 3D nano-helices have been realized with focused helium beam stress induced deformation technique, starting from a prefabricated cantilever on Al/Si_3_N_4_ film [104]. As compared to the Ga beam, the helium beam provided a larger control on the bidirectional folding angle (from −160° to +75°), and the possibility to shrink the external radius down to 100 nm.

#### 4.1.4. Nano-Kirigami

Another advanced technique capable to develop complex structures with broken symmetry, even in the form of metasurfaces exhibiting intrinsic chirality, is inspired by one of the most traditional Chinese art, the nano-kirigami [105]. Here, a thin film is irradiated by the beam in milling mode, first with a high ion dose and then with a low ion dose. The gallium ions implanted into the atomic lattice introduce a compressive stress, provoking the detachment of the structure from the surface. The stress can be topography-guided during the ion beam irradiation, in order to engineer fully 3D nanostructures. From a topological point of view, there are two main types of FIB nano-kirigami methods [106,107]: The tree-type folding/bending (Figure 5d), where the relative motions within each unit of the structure is separated from the others; and the close loop system, where the relative motions of the units interact with each other (Figure 5e).

These structures have been initially studied for their mechanical properties, but since 2015, their optics properties have been explored, too. Cui et al. [108] were the first to observe a FIB tree-type folding “nanograter” structure of vertical split ring resonators (SRRs) which exhibited sensitive Fano resonances in the NIR wavelength region. Other studies involved vertical plates at the place of SRRs [109,110] that also demonstrated strong plasmonic coupling and the appearance of pronounced Fano resonances arising by the grating structures. Particularly, the strong coupling modes are induced by the plasmonic resonances of the metallic holes and the ones generated along the edges of the vertical structure. However, with this technique it is only possible to develop structures with a regular shape. More complex metasurfaces also exhibit intrinsic chirality, such as gold 3D pinwheels (Figure 5f) developed with close-loop nano-kirigami [111] technique. The beam collides on the gold thin films, causing the sputtering of the gold atoms away from the surface, creating vacancies which stress the gold surface. At the same time, the gallium ions implanted into the film induce compressive stress, too. The combination of these two stresses creates the suspended gold film. To evolve in metallic pinwheel array, the basic principle is to rotate vertical helices in order to form horizontal cross-linked helices. Here, chiroptical effects were observed in the telecommunication wavelength range. The behavior of 2D and 3D pinwheel arrays with lattice periodicity of 1.45 μm was compared. The 3D pinwheel structure exhibits circular dichroism in transmission, significantly enhanced as compared with the one of the 2D counterpart (Figure 5g).

#### 4.1.5. Focused Ion Beam Milling for Stress-Induced 3D Chiral Fractal Metasurfaces

3D Archimedean spirals are another example of chiral structures (Figure 5h) which can be realized by the stress induced through FIB milling processing.

In this technique [112], the focused ion beam irradiates a freestanding metal/dielectric films. The 3D Archimedean spirals are stacked from a bilayer planar film of Au/Si_3_N_4_ (50 nm Au and 50 nm, respectively), as schematically shown in Figure 5i. The morphology of the final structures is related to the distortions induced by the beam conditions, and by the milling sequence, in combination with the material features and the layer thickness. Here, the beam irradiates the substrate along the spiral path, starting from the spiral center and introducing high stress and defects in the substrate. This causes the stretching of the milled spirals out of the plane with a height of around 800 nm. These metasurfaces exhibited highly enhanced chiral dissymmetry in transmission, with a broad operation band in the Mid IR range (Figure 5j), and, therefore, are promising for chiral sensing and vibrational circular dichroism spectroscopy.

### 4.2. 3D Chiral Nanostructures Realized by FIB-Induced Deposition

#### 4.2.1. 3D Chiral Nanohelix Growth

When scanning the ion/electron beam under opportune parameter conditions, in conjunction with the injected gas precursor, free-standing nanostructures can be grown with extremely high complexity, and even helix-base nanostructures, representing the ideal 3D chiral geometry, can be manufactured. Moreover, this technology offers unprecedented flexibility with respect to the complete set of nanohelices structural features (such as sizes, number of loops, and also number of intertwined wires), along with handedness, material composition, and spatial arrangement if inserted into an array configuration. Indeed, given its 3D nature, several geometrical parameters characterize the helix structure and drive its optical response: Namely, the vertical pitch (VP), the external diameter (ED), the wire diameter (WD) and, when arranged in ordered array configuration, the lattice period (LP) (Figure 6a).

Each of these parameters has an effect on the optical response, in both far and near field. First of all, the relationship between helix diameter and incident wavelength determines its operation in axial mode or in normal mode [113,114]. When the nanohelix features are comparable or smaller than the incident wavelength, plasmonic oscillations arise in the case of metal [115], and photonic stopbands [116] in the case of dielectric. In the case of metal, the plasmonic resonances observed in far field, as a function of circular polarized light, depend on the number and length of dipoles excited along the helix, and, therefore, are strictly related to VP and ED [117]. Hybridization phenomena [118] occur among the plasmonic resonances of multiloop systems, while in multiple systems such as arrays or multiwire helices, mutual helix or wire interactions can also affect the overall response. In-plane (LP) and out-of-plane (VP) periodicities are the main rulers in the case of dielectric helix-based photonic systems. Therefore, all the geometrical parameters play a relevant role in assessing the behavior of a helix-based system. Moreover, in this case, as for all the other chiral shapes, any deviation from expected geometric or structural value heavily affects the response in circular dichroism experiments, giving nanofabrication accuracy an extremely critical role.

In the last years, our group extensively worked on the FIBID process capability in the manufacturing of helix shaped metamaterials, demonstrating a high level of accuracy and uniformity, starting from the single elements to array configurations. For a controlled growth in the third dimension, the relationship between the beam energy and the step size must be considered: Starting from an empty circle design, the growth in the third dimension is, at a first stage, ruled by the ion energy and the step size, which controls the overlapping of nucleation sites. At low ion energy, the structure is not able to detach from the substrate, while, on the other hand, a too-high ion dose promotes extra evolution in the third dimension for small step size, or an extra seeding point for large step size (Figure 6b). Another relevant parameter for helix growth is the dwell time, which defines the amount of the deposited material in each spot (i.e., the vertical growth rate) and can be set to control the evolution of each loop in relationship with the step size, as schematically represented in Figure 6c. Additionally, since the local pressure of the gas precursor affects the amount of deposited material, to overcome the pressure gradient in the vacuum chamber, a progressive increase of the dwell time [119] has been demonstrated to increase the nucleation sites during the process, defining a continuous vertical growth.

As discussed in Section 2, along with the interaction between the ion beam, the substrate and the precursor molecules scattering, proximity and charge effects also arise from the interaction of the beam with the growing structure, in the case of this complex helix shape, when multiple loops are grown. The proximity effects, together with the variation of the local pressure, are responsible for the growth rate and the spiral diameter variation. If the dose is kept constant, the size of the vertical pitch decreases with the number of loops along the *z*-axis (Figure 6d,e). Thus, to obtain precise nanostructured helices with constant vertical pitch, a dose compensation protocol needs to be applied [54]. The minimum WD achievable with the Ga beam is 80nm, limited by the instrumental resolution. Further shrinkage can be obtained by using the electron beam (WR ~40 nm) [69,120] or focused helium beam (WR ~45 nm) [121].

Recently, the accuracy of FIBID in nanohelix manufacturing allowed Wang et al. to demonstrate an innovative scheme of [88] subwavelength polarization optics. The proposed system integrates a nanospiral in near field with a rectangular aperture nanoantenna. The spiral is engineered to operate in the nonresonant axial mode at telecommunications frequencies: The helix core (1.66 μm height, with external diameter of 505 nm) is a carbon wire with a diameter of 105 nm fabricated by FIBID, and then sputter coated with a gold shell of 25 nm. The rectangle nano-aperture is engraved by FIB milling in the flat gold layer at the helix pedestal. The so-built nanoantenna generates a background-free directional light beam with a geometrically tunable degree of circular polarization and ellipticity factor (Figure 6f).

Moreover, by closely positioning four nanoantennas, a more complex polarization response can be obtained, because of the plasmon coupling between helices of different handedness, resulting into a subwavelength waveplate-like structure. Such local sources of CPL can have application in photonic information processing, polarimetry, miniaturized displays, optomagnetic data storage, microscopy, sensing, and communications.

The same group also proposed the successful integration of a single nanohelix at the apex of a sharp tip used in scanning near-field microscopies [122]. Figure 6g shows the dielectric near-field tip covered with an Al layer of 100 nm and used as a substrate for fabrication of a carbon-gold core-shell helix by FIBID and sputter-coating. Then a nanoaperture has been milled in the tip metal-coating. This fabrication method can be employed for any tips, also with nanometer range apex like atomic force microscopy probes to realize moveable, broadband, and background-free chiroptical probes.

#### 4.2.2. Optical Properties of Nano-Helices Arrays

Because of its local nature, FIBID allows us to create arrays of helices, precisely positioned in a predefined lattice. In this case, the growth is complicated by proximity effects related to the presence of a nearby structure, and by local pressure variations. The flow conditions can be properly optimized by setting the GIS position with respect to the writing zone and the distance from the substrate, followed by the application of dose compensation strategy.

These conditions proved to be effective for small arrays of 5 × 5 elements. For larger array sizes, the local pressure reduction becomes significant, and to restore the pressure in the chamber, a refresh time should be periodically introduced. This solution provided accurate size control for arrays containing up to 20 × 20 elements, which is the largest number of helices made using this technique (Figure 7d) [69]. However, as the number of elements increases, the control of all these parameters is more difficult, leading to lower accuracy and size uniformity. This limits the application of the process to wafer-scale sizes and requires optical measurements in confocal microscopy geometry for spatial selection.

We studied the optical behavior of nano-helices arrays as a function of the loop number in two different material systems, and dielectric and metal helices. The transmission spectra of left- and right-handed circularly polarized light give rise to two dichroic bands in the range of the visible thanks to the chosen geometrical parameters (Figure 7a).

In the case of platinum helices [81] the fundamental unit (meta-atom) that starts exhibiting CD in transmission is found for half loop metal helices, and this effect increases for further increases of the loop number, as shown in the transmission spectra of Figure 7a. Moreover, a red shift of the spectral features has been observed with the increasing number of loops, because of plasmonic resonance hybridization phenomena. For arrays with larger N (SEM image in Figure 7d), the combination of high refractive index contrast medium with 3D nanoscale arrangement of chiral structures, allowed for a difference in CPL transmission of 20% in the VIS and beyond 36% in the NIR [69]. For the optical activity (the rotation angle of linearly polarized incident light), the spectra in Figure 7c measured for the same arrays employed to record the transmission spectra of Figure 7a show maximum values of around 4.5° and 3° at 575 and 850 nm, respectively. Therefore, this system operates in a broadband spectral region, as required for applications like integrated super achromatic optical rotators. Pt helix arrays were also studied through photoacoustic measurements, to obtain the CD of the chiral sample related only to absorption, thus avoiding interferences from diffraction effects [123].

As shown in the previous section, chiral nano-helices grown with FIBID with a carbon-based precursor have a structural composition that can be classified as low-k dielectric. Their optical response indicated a dielectric behavior since the difference among LCP and RCP curves leads to two circular dichroic bands induced by the matching of the light wavelength with the structure vertical pitch. The transmission spectra intensity decreases with the number of loops (N) from N = 0,5 to N = 1,5, but the two dichroic bands for each structure are higher than the Pt ones with the same dimensions, because of lower material losses.

#### 4.2.3. Multiple Nanowire

As we have already seen, 3D helices can be considered as the best candidates for pronounced chiral response integrated in compact devices. However, they suffer from the loss of rotational symmetry and of a residual linear birefringence, which limits the polarization conversion purity, and provides high sensitivity of the optical rotation to the structure orientation.

To overcome these problems, additional levels of complexity and different functionalities could be added to helix-base metamaterials through different spatial arrangements, which can mean compact arrays or alternative designs based on multiple wire systems. For example, rotational symmetry can be effectively restored with a multi-helical nanowire (MHN) configuration, as numerically proposed by [124].

FIBID processing of helix metamaterials allows solutions for on demand manufacturing and full control on all these features, with effects on the operation bandwidth, transmitted polarization purity, and circular/linear birefringence. Not only compact or loose configurations, in plane and out of plane, can be tailored by FIBID technology, but also branched configurations are accessible. FIBID extrusion in the third dimension of MHN is hindered by the presence of blind spots (the space in which the already grown structures lies). In addition, the proximity effects of the single wires can limit the achievable size accuracy. To overcome all these difficulties, a tomographic rotatory strategy has been developed [125]. Here, the growth of each wire is divided in multiple arches and combined with a split circular beam scan. The method can be applied for N-wires depending only on the resolution of FIBID technique, whereas smaller radius can be obtained with FEBID processing. Figure 7e shows a SEM image of triple helical nanowires (THNs) array arranged into a compact square lattice array of 10 × 10 µm elements with *LP* = 700 nm.

#### 4.2.4. Influence of the Helix Array Spatial Arrangement

The impact of spatial arrangement enabled by FIBID technology flexibility, can be seen in Figure 7f to k where three different helix layouts are compared with respect to g factor, signal to noise ratio (SNR) and optical rotation dispersion as a function of sample orientation [126]. The designs are: Two arrays with *LP* = 900 nm and *N* = 1, of single wire and multiple wires helices, respectively: And one array of closely coupled single nanowire helices with *LP* = 700 nm and *N* = 3.

Figure 7i shows the comparison between the dissymmetry factor *g*, defined as *g* = 2(*T*_LCP_ − *T*_RCP_)/(*T*_LCP_ + *T*_RCP_) where *T*_LCP_ and *T*_RCP_ are, respectively, the left and right-handed component of the circularly polarized transmission light.

The single helical nanowire array with *LP* = 900 nm shows a null g-factor, while the most closely-packed geometry displays a maximum g factor of 25%, due to both the higher number of loops and higher light-matter interaction volume. For the THN system, instead, a further increase of the g factor up to 70% is observed, together with larger enhancement in the CD and in bandwidth.

The SNR parameter (defined as logarithm of the ratio between the two transmitted circularly polarized light components) provides information about the purity of the circularly polarized light (Figure 7j). It was found that for the single nanowire arrays configuration, the SNR value does not reach 10dB. In the THN array case, a maximum value of 25 dB between 500–600 nm is observed, thanks to internal coupling among each single wire, which also causes the bandwidth enlargement throughout the whole VIS range.

For the most compact array, the mutual helix plasmonic interaction is boosted, resulting in a higher g factor and larger bandwidth than the other SHN configuration in which they are further away. However, this closer arrangement causes a decrease of the averaged transmission intensity, thus reducing potential device efficiency. In the THN arrangement, instead, the helices are internally excited by the incident light on each nanowire with a 3-fold symmetric current path which improves the SNR.

Finally, optical rotatory dispersion analysis (ORD) using linearly polarized incident light was performed. In helical metamaterials, the optical rotations arise from the excitation of electric and magnetic dipoles along the vertical axis, due to two different refractive indices for the two circular polarizations. In single wire helical-based metamaterials, the rotational symmetry is not preserved because of a preferred direction in space determined by the tip at the ending part of the helix together with the helix axis [124] that induces intrinsic linear birefringence. Figure 7k displays the polarization rotation angle dependence on the sample orientation measured at the wavelength with null circular dichroism, in order to detect exclusively the pure linear polarization rotation. The array system with larger LP displays an optical rotatory dispersion with linear birefringence of 0.8° with an ORD mean value below 1°, because of reduced external coupling. Higher ORD modulation is recorded for the closely packed single-wire nano-helices array with a mean value of 7°, maximum optical rotatory dispersion of 12° and strong modulation with an excursion up to 8°. Conversely, the THNs array shows a constant ORD value of 8° in a broad wavelength range, with a very faint modulation of less than 0.5° (related to fabrication tolerance), highlighting the weak linear birefringence due both to the recovery of rotational symmetry, and to the strong internal interaction among the wires which overcome the external coupling.

These results promote THN as the configuration to achieve a strong and very uniform polarization rotation with respect to the sample orientation, and proved the critical role interplayed by the chiral properties and the metamaterial design for 3D nanoscale optical devices engineering.

## 5. Conclusions

We have seen the recent progress in the field of nanofabrication of 3D nanostructures and metamaterials made by FIB processing. The implementation of both FIBID bottom-up and milling top-down technologies allowed us to realize chiral structures with a broadband operation range. We have observed the strategies adopted to obtain 3D complex chiral nanostructures (single and periodic) and metasurfaces with an accurate control of the nanoscale geometry to develop any kind of photonic structures ranging from photonic crystals to metamaterials.

Large chiroptical effects have been observed from chiral nanostructures made with both techniques, since the third dimension is an important feature when exploiting intrinsic chirality. All the strategies observed have shown tunable and broadband optical chirality ranging from visible to infrared (IR). The chiral geometries of plasmonic nanostructures currently developed with these technologies can effectively enhance the light-matter interaction and increase the otherwise weak signal of molecules with potential application in biology and chemistry. Each presented study involves different applications, in several technological fields like nanophotonic devices, biosensors, and for high resolution imaging.

In what follows, we make a short comparison with other methods currently available for the fabrication of 3D chiral nanostructures or metasurfaces at the nanoscale: Glancing angle deposition (GLAD) [127,128,129,130,131,132,133,134,135], multi-step electron beam lithography (EBL) [136,137,138,139,140,141], DNA Self-assembly [142,143,144,145,146], nanoimprint lithography (NIL) [147,148], and direct laser writing (DLW) [149,150,151,152,153,154].

### 5.1. Glancing Angle Deposition

It is based on a deposition process with a multiple step approach. The template is formed by a non-homogeneous nucleation followed by physical vapor deposition under an oblique angle on a rotating substrate. A former lithography procedure helps to realize the seeds tailoring the growth conditions for obtaining the required geometric features.

Glancing angle deposition demonstrated the ability to produce large-area arrays of helix-shaped nanoparticles with feature sizes of the order of a tenth of nanometers, and with fabrication speed higher than DLW and FIBID/FEBID techniques. However, achievable shape complexity is limited.

GLAD provides a large material choice, varying from metals to dielectrics exploited to modulate the chiro-optical response from visible to UV [129,130,131,132]. Recently, for example, chiral nanostructures in refractory material (titanium nitride) arranged in a regular large-area array and dispersed in solution have been studied [127]. The measured chiro-optical response is demonstrated to be very broadband (from 500 to 1400 nm) with contributions from individual and collective plasmon modes. Core–shell nanostructures manufactured with this technology exhibited strong chiro-optical activity with spectral characteristics tunable with the shell thickness, to obtain the optical response from visible to near IR [128].

GLAD chiral nano-helices have been employed for chiral sensing with high sensitivities and figures of merit, engineering the dispersion function of these structures to be optically active in VIS (with gold) [133] and for large chiro-optical effects in the UV (with TiO_2_ [134] and Mg [135]).

### 5.2. Multi-Step Electron Beam Lithography

It utilizes a focused electron beam to write a nanoscale pattern which is then processed in subsequent steps, also involving metal deposition and lift-off. The 3D structure is then obtained by repeating the stacking of 2D metamaterial layers on top of each other. Given the small size of the electron probe, the resolution is very high (<10 nm), and it is scalable from small to large area fabrication. The simplest structure that can be done by this technique is an array of two orthogonally coupled nanorods. In ref. [136], the authors demonstrated that this system can follow the Born-Kunh model, since it can present symmetric or antisymmetric modes associated with distinct handedness. The tuning of the vertical distance between the rods lead to selective excitation by RCP or LCP light. Twisted optical metamaterials also demonstrated to discriminate between two enantiomeric forms present in a solution with zeptomolar concentration [137]. Other complicated forms have been used with this technique, such as split-ring resonators [138] and crosses [139].

Another strategy, called on-edge lithography, starts from transferring a design to the substrate via wet etching or DLW. Then EBL, metal evaporation, and lift-off are applied to get 3D metallic nanostructures. With this technique, L-shaped structures on silica substrate were developed building first long trenches, on which the structures have been fabricated by the following EBL steps [140]. Large CD was observed in the range from 800 to 1400 nm. Metallic nanostructures with fourfold rotational symmetry have also been realized with such a technology, to avoid the problem of the linear birefringence caused by the absence of rotational symmetry of L-shaped structures [141].

### 5.3. Self-Assembly

It consists of a top-down approach involving chemical synthesis: By combining different materials and strategies, it is possible to obtain 3D colloidal chiral nanoparticles dispersed in solution.

In particular, the DNA self-assembly can use DNA origami and DNA scaffolding. In the former, single-stranded DNA (ssDNA) are used to create controllable binding sites for the functionalized nanoparticles, leading to the assembly of nanoparticles in a helical shape. The latter exploits a single ssDNA strand on individual nanoparticles to bind two or more selected nanoparticles following a designed scheme. Both methods demonstrated a controlled spatial configuration, flexibility in size, suitability for large volume production, and programmability.

With the origami method [142], gold helical-like nanoparticles with sizes smaller than 60 nm and exhibiting optical response in visible were realized. It was also demonstrated that, by surrounding the nanoparticles with a silver shell, the CD can be further enhanced. In another work, gold nanoparticles arranged in two origami sheets were subsequently rolled to form a spiral, and chiroptical response was observed at 550 nm, close to the single nanoparticle resonances [143]. Beyond nanoparticles, stacks of rotated gold nano-rods can also be assembled to form helicoidal shapes using DNA nanosheets, exhibiting a chiral optical response at around 700 nm in the mdeg range [144]. Exploiting DNA assembly, even other complex chiral shapes can be built like nanoparticle pyramids [145] realized with two gold nanoparticles with different sizes, a silver nanoparticle and a quantum dot. Precise control over the position and sequence of the nanoparticles has been demonstrated. Four identically-sized gold nanoparticles were employed to form chiral L-shape quadrumers [146], with handedness depending on the position of the fourth particle relative to the L-shape. Large chiral optical response due to the efficient plasmonic coupling of the particles was reported, together with the observation of chiral features present in the whole configuration.

### 5.4. Nanoimprint Lithography

It is based on the transferring of a designed master mold into a resist, with a resolution of 2 nm. It is available for micro/nanostructures with both high throughput and low cost. However, the processing difficulty increases with the complexity of the 3D structure, where the combination with other techniques is necessary, limiting the fabrication resolution. As an example, in Ref. [147], 3D micro- and nanostructures have been developed by combining nanoimprint lithography and optical lithography by direct imprinting of thin-film of metal 2D patterns formed on a polymer. However, the optical response is observed in the frequency range of 1.7–9 THz. On the other hand, large-area (cm-scale) 3D L-shaped chiral plasmonic substrates [148] have been realized using a combination of NIL and GLAD. These consist of two layers of Au nanorods separated by a Ge dielectric layer orthogonally oriented between them. They have shown strong optical chirality with a g-factor up to 0.38 in IR region.

### 5.5. Direct Laser Writing

It is based on femtosecond laser pulses focused into a photoresist. A computer-controlled scanning of the focus with piezoelectric actuators induces photo-polymerization in order to fabricate complex 3D and chiral structures. As FIBID, it is limited by the low writing speed. This technique does not allow us to scale down to the nanometer sizes (the maximum lateral resolution achievable is 500 nm), owed to the resolution limit that confines the optical response in the Near/MID-IR. Moreover, to study chiral plasmonic effects in metallic structures, a second step of metal deposition by electroplating is needed.

Two-photon direct laser writing followed by a gold electroplating step has been employed by Gansel et al. to build the first example of periodic arrays of chiral helices with plasmonic modes along the whole wire acting as a broadband circular polarizer [149]. They also demonstrated more complex configurations such as tapered [150], double handedness [151] and, in combination between stimulated emission depletion (STED) microscopy inspired DLW, multiple wire helices [152] with improved functionality in terms of polarization conversion in THz and IR range.

3D chiral photonic crystals have also been developed with this technique. In [116], polarization stop bands with large circular dichroism in transmission in IR have been observed from high-quality 3D polymeric helices arranged in periodic array.

In this direction, even most complicated shapes like array of bi-chiral dielectric photonic crystals (PhC) with cubic symmetry can be realized with pronounced polarization stop bands in IR [153]. Recently, Liu et al. have been able to shrink the lattice constants of chiral photonic crystals in order to act in the UV–visible spectral range (100–700 nm). The technique employed in this work combines the two-photon polymerization lithography (TPL) and STED with a heating procedure reducing the PhC period down to 280 nm. This process demonstrated the possibility to create patterns of colors depending on the photonic crystals size [154].

FIB processing still requires huge efforts to solve the issue of material complexity, to increase fabrication speed, and to extend the fabrication area for wafer-scale implementation. We have seen how the impurities caused by milling can benefit the fabrication in the third dimension, as well as how the unavoidable fabrication defects help to improve the chiro-optical response of the milled surfaces. For better accuracy, focused helium beam milling can be performed, to avoid material redeposition and to refine with higher precision of the edges. Moreover, the smaller ion source can help in induced deposition to further shrink the minimum size of the structures, providing further degrees of freedom in the tunability of the optical response.

New frontiers of technology arise from the cryogenic temperatures, which provide higher resolution and reduced processing time, preventing the side effects caused by ion irradiation.

Another important material strategy can take advantage of the direct employment of the constituent ions of the beam source. Complex chiral plasmonic systems based on Ga nanostructures can be created by the interplay between the beam parameters and host media (the precursor) with a low molecular weight, leading to chiral plasmonic structures. Indeed, new research is currently under development for FIB source technology, investigating new kinds of alloy sources [155], even noble metals source like gold [156], always considered as the prime choice in plasmonic. Complex plasmonic structures with superior optical properties and functionalities can be realized with these innovative sources.

Finally, to the aim of a broad application gamut, the target of large area scalability is needed; a possible strategy could be the integration of a laser interferometer stage [39,156].

These new and promising technological tools and the continuous progress driven by the research community towards miniaturized devices with increasing design complexity, envision novel possible chiral functionalities that can be integrated in nanophotonic systems for practical applications by using FIB processing.

## Figures and Tables

**Figure 1 micromachines-12-00006-f001:**
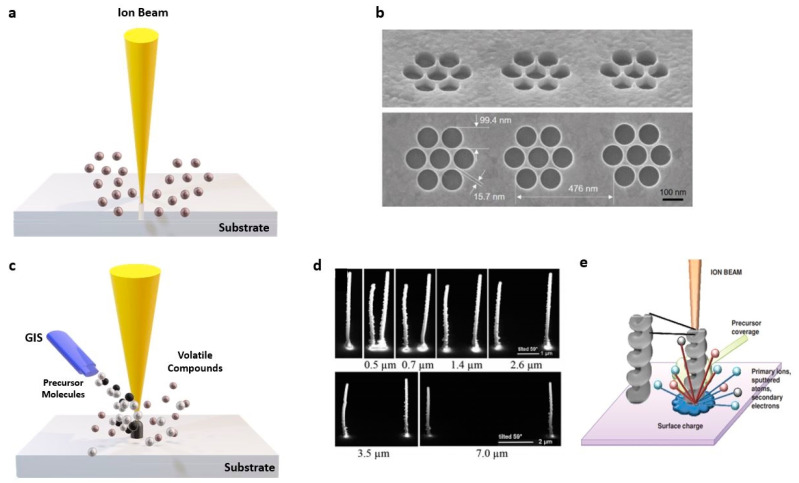
(**a**) Scheme of the focused ion beam milling. The ions coming from the beam collide on the surface, digging the targeted area. (**b**) Helium Ion Microscope (HIM) images of 54° tilted view (top) and top view (bottom) of plasmonic nanohole arrays realized with He^+^ milling on Au/SiO_2_ substrate. The array consists of seven close-packed holes with 100nm diameter and separated by 15nm of sharp edges without milled material redeposition. Reproduced with permission from [48]. (**c**) Scheme of the focused ion beam induced deposition (FIBID) procedure: The ions coming from the beam decompose the precursor molecules coming from the gas injection systems leaving a deposit on the substrate. The residual organic compounds are removed from the vacuum system. (**d**) Scanning electron microscope (SEM) images representing the proximity effect for six pillar pairs with different separations gaps from 0.5 up to 7.0 μm. The metal-organic precursor gas (CH_3_)_3_Pt(CpCH_3_) is used as the gas source. The pillars have been grown on a Si_3_N_4_/Si wafer. Reproduced with permission from [58]. (**e**) Schematic image representing all the effects happening during the FIBID process in the fabrication of a multiple loops nanohelix: The interaction with the sample led to scattered particles, charge effects, and proximity effects. Reproduced with permission from [54].

**Figure 2 micromachines-12-00006-f002:**
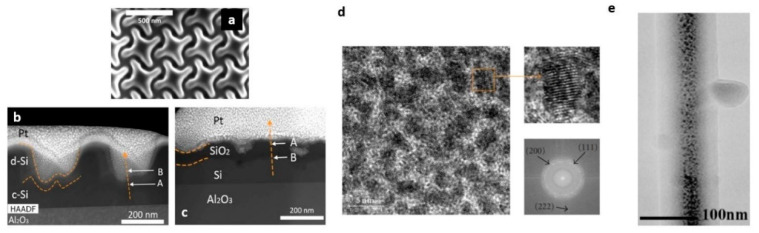
(**a**) Top view of FIB-milled 3D chiral photonics silicon nanostructures on a single crystal silicon film on sapphire. The period of the nanostructures was set to 370 nm. (**b**,**c**) Scanning transmission electron microscopy (STEM) cross-section images of the nanostructures before and after the annealing. A previous coating of a layer of platinum deposited in two stages has been applied as a protective layer. The STEM image taken before the annealing (**b**) shows: The presence of gallium implanted on the layer of damaged silicon (d-Si in the image) above the surface of the nanostructure, amorphous silicon inclusions (c-Si in the image), and the deposited protective platinum layer. The STEM image taken after the annealing shows that the d-Si is removed and a layer of silicon oxide is formed. Reproduced with permission from [68]. (**d**) HRTEM images of deposited Pt films grown on a copper grid show bright and dark regions associated to amorphous C region of carbon matrix and Pt grains, respectively. The inset shows the magnification of a Pt grain in order to observe the atomic planes, while the image below represents the diffraction spots of the Pt grains that correspond to the (200), (111), (222), and (202) atomic planes of fcc. Reproduced with permission from [76]. (**e**) TEM of an amorphous diamond like carbon (DLC) pillar. One can see the dark contrast of nanograins from the gallium core, while the bright part is the thick carbon shell, classified as diamond like carbon. Reproduced with permission from [80].

**Figure 3 micromachines-12-00006-f003:**
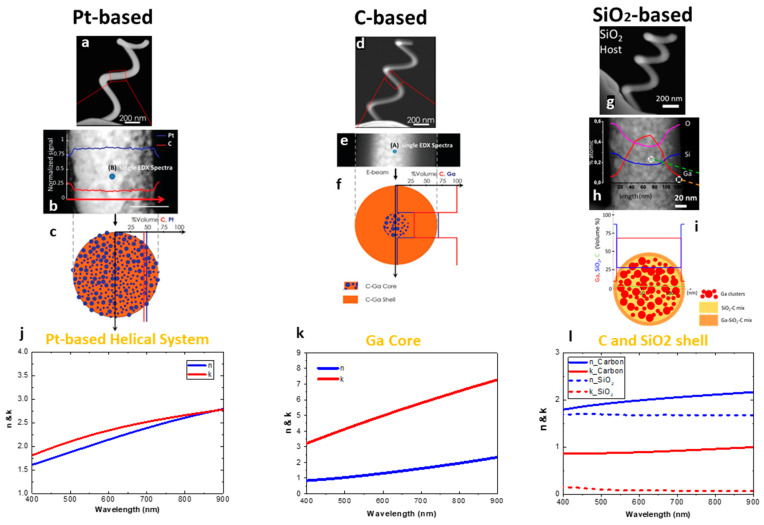
(**a**) STEM in HAADF (High Angle Annular Dark-field) image of a nanohelix realized with FIBID using Pt-precursor. (**b**) High-magnification STEM-HAADF image of a section of the Pt nanohelix. (**c**) Schematic view of the wire cross section which shows that the structure is composed by Pt nanograins embedded in an amorphous C matrix. The blue points correspond to Pt grains, while the amorphous carbon matrix is indicated in orange. (**d**) STEM in HAADF image of a nanohelix realized with FIBID using C-precursor. (**e**) High-magnification STEM-HAADF image of a section of the c nanohelix. (**f**) Scheme of the wire cross section of C nanowire made of Ga precipitates (diameter 1.5–4 nm) incorporated within a 30 nm core and an outer carbon shell. The blue spots indicate the gallium insert, while the orange indicates the carbon matrix. Reproduced with permission from [81]. Copyright © 2016, American Chemical Society. (**g**) STEM in HAADF image of a nanohelix realized with FIBID using SiO_2_-precursor. (**h**) High-magnification STEM-HAADF image of a section of the SiO_2_ nanohelix. (**i**) Scheme of the wire cross section of SiO_2_ nanowire, which displays Ga nanoparticles (with averaged diameter of 12 nm) and the outer SiO_2_ shell. The red spots indicate the gallium insert, the yellow sites indicate the SiO_2_ and carbon matrix, while the orange indicates the outer SiO_2_ shell. Reproduced with permission from [82]. (**j**–**l**) Analytical dispersion values (n-blue line, k-red line) for the Pt-based (**j**), the core-shell C-based (**k**), and the SiO_2_-based (**l**) nano-helices retrieved by FTDT simulations.

**Figure 4 micromachines-12-00006-f004:**
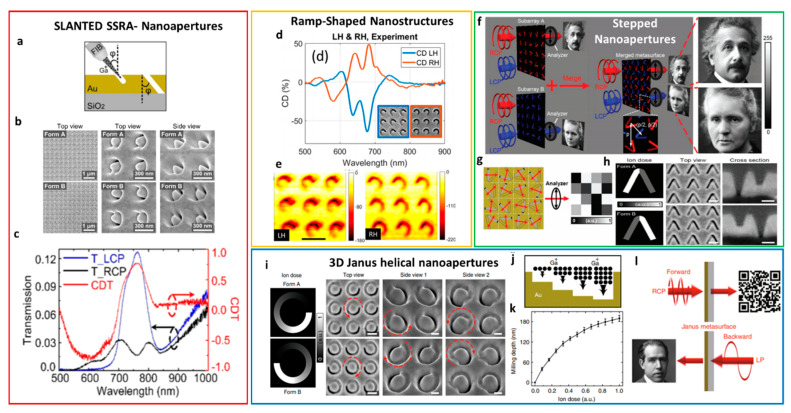
(**a**) Sketch of the tilted angle focused ion beam (FIB) process. (**b**) From left to right: SEM images of the slanted split ring apertures (SSRA) chiral metamaterials array in the two enantiomeric forms A and B; SEM images of the unit cell SSRA chiral metamaterials array in the two enantiomeric forms A and B; side view of SSRA chiral metamaterials realized with a slant angle of 40° in the two enantiomeric forms. (**c**) Measured transmission spectra of the SSRA chiral metasurfaces under LCP and RCP (L_LCP blue line and R_RCP, black line respectively) and the circular dichroism spectra (CDT, yellow line) calculated as CDT = (TLCP − TRCP)/(TLCP + TRCP). Reprinted with permission from [92]. Copyright © 2018, American Chemical Society. (**d**) Comparison between the measured circular dichroism of left (blue-line) and right (orange line)– handed nanostructures calculated as CD = L − R. The inset represents the top view of the SEM image of the ramp-shaped left and right-handed nanostructures. The structures have 100 nm of radius, a thickness of 200 nm and the period of the array is 600 nm. (**e**) Atomic force microscopy (AFM) images of the left-handed (LH) and the right-handed (RH) structures. The scale bar is 600 nm. In the images the darker pixels correspond to the deeper points. Reprinted with permission from [96]. Copyright © 2019, American Chemical Society. (**f**) Chiral grayscale subwavelength imaging design. The subarrays A and B are composed by two opposite stepped V-shaped nanoapertures enantiomeric forms. When interacting with circularly polarized light (CPL), if an analyzer is inserted after the metasurfaces to form the grayscale images, the subarrays form A can generate the grayscale portrait of Einstein while the form B can generate the grayscale portrait images of Marie Curie. (**g**) Schematic of grayscale intensity profile generated by an array of stepped V-shaped nanoapertures. The scale bar is 500nm. (**h**) Ion dose distribution and SEM images of the two enantiomeric forms of stepped V-shaped nanoapertures fabricated using the grayscale focused ion beam milling method. The cross-section images are acquired with a tilted angle of 42°. The scale bar is 200nm for the top view and 50 nm for the cross section. Reused with permission from [97]. (**i**) Ion dose distributions of the two enantiomeric forms of Janus 3D metasurfaces and SEM images of the fabricated 3D helical nanoapertures. The scale bar is 200nm. The side-view images are acquired with a tilted angle of 52°. The dashed lines indicate the growth direction of the groove. The scale bare is 100nm. (**j**) Schematic illustration of the ion dose distribution of grayscale focused ion beam milling method. (**k**) Graph of the milling depth as a function of the applied ion dose. (**l**) Janus metasurfaces work principle: In the forward direction, the right circularly polarized light is transmitted and displays a QR code, while in the backward direction, linearly polarized light gives back a grayscale image. Reprinted with permission from [94].

**Figure 5 micromachines-12-00006-f005:**
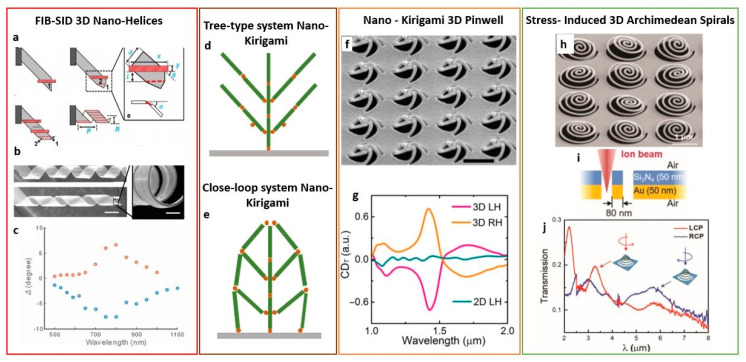
Scheme of the fabrication steps of the focused-ion-beam stress-induced deformation (FIB-SID) 3D helix. The aluminum strip is irradiated from the tip by the ion with different ion doses which define the bending angle. By repeating the procedure different times, the 2D strip can be curled into the 3D helix. The subfigure (**a**) shows the fabrication parameters: α and θ are, respectively, the folding and bending angles, t is the folding step number, and l is the step period. (**b**) From left to right: SEM images of two nano-helices fabricated different handedness. Zoom of the top side of a fabricated FIB-SID helix. The scale bars are 2 µm and 500 nm. (**c**) Optical rotation versus wavelength curves measured for two chiral nanohelices with same radius of *R* = 1.2 µm and two different handedness. Reproduced with permission from [103]. (**d**,**e**) Schematic representation of the topological classifications of multibody systems: (**d**) Tree-type system where the motion of each structure units (green line) is not connected between each other. (**e**) Close-loop system. The relative motions of the structure units interact with each other. The connectors between the units are represented by the orange circles. (**f**) SEM image of left-handed (LH) pinwheel arrays. Scale bar: 1 μm. The arrays have periodicity of 1.45 mm. The height is about 380 nm. (**g**) Measured circular dichroism spectra for 2D and 3D LH and 3D RH pinwheels. Reproduced with permission from [111]. (**h**) Side view of SEM images of the 3D Archimedean spirals metasurfaces. (**i**) Sketch of the cross-section of the Au/Si_3_N_4_ bilayer under FIB irradiation. (**j**) RCP (blue line) and LCP (red line) transmission spectra of the 3D Archimedean spirals. Reproduced with permission from [112].

**Figure 6 micromachines-12-00006-f006:**
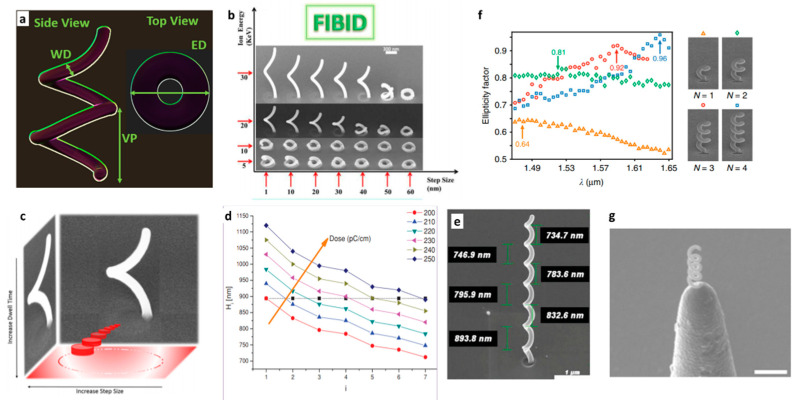
(**a**) Scheme of the helix geometrical parameters: Vertical Period (VP), External Diameter (ED), the wire Diameter (WD). (**b**) Scheme of the nanohelix evolution in the third dimension as a function of ion energy and step size. Reproduced with permission from [69]. (**c**) Scheme of the helix growth balancing the step size and the dwell time. Reproduced with permission from [119]. (**d**) Evolution of the vertical pitch during the multiple loop helix growth. The dashed line represents the design VP. (**e**) SEM image of a Pt-based nano-helix with seven loops realized by FIBID. The vertical pitch (VP) decreases along the *z*-axis from 893 nm down to 735 nm, as a result of proximity effects. d-e reproduced with permission from [54]. (**f**) Spectrum of the ellipticity factor of the single HTN emission measured for different turns: One turn (orange triangles), two turns (green diamonds), three turns (red circles), and four turns (blue squares). Reproduced with permission from [88]**.** (**g**) SEM image of an antenna grown at the apex of a tip used in scanning near-field microscope. The scale bar is 1 µm. Adapted and reprinted with permission from [122], © The Optical Society.

**Figure 7 micromachines-12-00006-f007:**
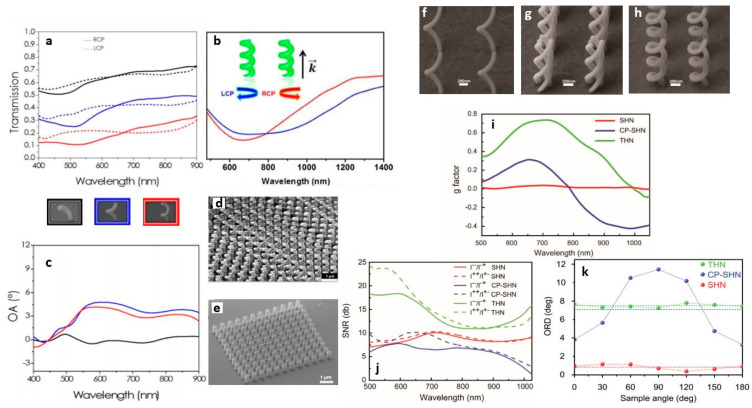
(**a**) Experimentally measured LCP and RCP transmission curves for arrays of nanohelices Pt-based (**a**) with *N* = 0.5, 1, 1.5 loops (black, blue, and red lines, respectively) and size *ED* = 300 nm, *WD* = 100 nm, *LP* = 500 nm, and *VP* = 500 nm. Reproduced with permission from [81]. (**b**) Measured transmission spectra of right- and left-handed circularly polarized light (red and blue lines, respectively) at normal incidence, of the array in Figure 7d. Reproduced with permission from [69]. (**c**) Measured optical activity for Pt-based helices arrays as in (**a**). Reproduced with permission from [81]. (**d**) SEM image of an array of 20 × 20 nanohelices (5 loops, *ED* = 400 nm, *WD* = 130 nm, *VP* = 300 nm, *LP* = 700 nm) fabricated by FIBID. Reproduced with permission from [69]. (**e**) SEM images of the THN array with sizes: *ED* = 375 nm, *WD* = 110 nm, and *VP* = 705 nm, *LP* = 700 nm. Reprinted with permission from [125]. (**f**–**h**) SEM image of SHN single loop (**f**), THN single loop (**g**), and closely packet SHN triple loop (**h**) arrays grown by FIBID of platinum on the ITO/glass substrate. WD and ED are kept constant at 110 nm and 380 nm, respectively, in the three samples. (**i**) g factor of the d, e, f helices (red line, green line, and blue line, respectively). (**j**) SNR calculated for f, g, h helices shown in Figure 7 (red line, green line, and blue line, respectively). The different components of transmitted light are I++, I+− and I−−, I −+, where + refers to LCP and − to RCP light. The first index identifies the incident circular polarization, while the second index corresponds to the detected polarization components. (**k**) Optical activity of the crossing point transmission values for the arrays of Figure 7f–h. Reproduced with permission from [126].

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
