# Peer review of "Focused Ion Beam Processing for 3D Chiral Photonics Nanostructures"

_micromachines, 2020, doi:10.3390/mi12010006_

Round 1

Reviewer 1 Report

In this manuscript, the authors presented a review of recent progress in FIB nanofabrication of 3D chiral structures for photonic applications. The manuscript mainly covers FIB milling and FIB deposition techniques for nanofabrication, and compares the advantages and disadvantages of the two techniques. The manuscript also includes discussions on the material composition issues and purification approaches of FIB nanofabrication. The manuscript reviews several FIB processing techniques for fabricating 3D chiral-optical nanostructures. This review is important and timely given the recent development in the fabrication of nanophotonic structures and devices based on FIB techniques. The manuscript is well organized, and the language and style are of decent quality. Adequate references are also included in the manuscript. I recommend acceptance of the manuscript, after the authors address the following comments.

About the title: this review primarily focuses on FIB nanofabrication for photonic applications. However, this is not fully reflected in the title. I suggest adding words including “optical” or “photonic” (e.g. “chiral-optical” or “photonic applications”) to make the title a more accurate and specific description of the research been reviewed.

This review includes research on FIB deposition and FIB milling. Another FIB-related technique is FIB-based charged-particle lithography, which uses the ion beam to exposure resist in a similar fashion to electron beam lithography. Even though FIB-lithography may not be a mainstream FIB technique, a review paper should briefly mention it and point interested readers to relevant papers.

Some chemical formulas (such as (CH3)3Pt(CpCH)3, SiO2, and Si3N4 in the caption of Fig. 1) need correct subscription format.

The figure quality of Fig. 3 is poor compared to other figures.

It is worthwhile to compare the throughput (fabrication speed) of FIB milling and FIB deposition, as this is import for fabricating large area devices and metamaterials.

Figure 6 is mentioned before figure 5 in the text. Consider swapping Figs. 5 and 6, or swapping the corresponding sections in the text, or re-arranging the panels in Figs. 5 and 6.

The labels of Figs. 6f, 6g, 6h are inconsistent with the description in the caption.

Author Response

In this manuscript, the authors presented a review of recent progress in FIB nanofabrication of 3D chiral structures for photonic applications. The manuscript mainly covers FIB milling and FIB deposition techniques for nanofabrication, and compares the advantages and disadvantages of the two techniques. The manuscript also includes discussions on the material composition issues and purification approaches of FIB nanofabrication. The manuscript reviews several FIB processing techniques for fabricating 3D chiral-optical nanostructures. This review is important and timely given the recent development in the fabrication of nanophotonic structures and devices based on FIB techniques. The manuscript is well organized, and the language and style are of decent quality. Adequate references are also included in the manuscript. I recommend acceptance of the manuscript, after the authors address the following comments.

1.About the title: this review primarily focuses on FIB nanofabrication for photonic applications. However, this is not fully reflected in the title. I suggest adding words including “optical” or “photonic” (e.g. “chiral-optical” or “photonic applications”) to make the title a more accurate and specific description of the research been reviewed.

We are grateful to the Referee for raising this point. Indeed, the review describes chiral 3D nanostructures that exhibit intrinsic chiro-optical properties thanks to design flexibility provided by FIB processing. Thus, we changed the title in “Focused Ion Beam Processing for 3D Chiral Photonics Nanostructures”

2.This review includes research on FIB deposition and FIB milling. Another FIB-related technique is FIB-based charged-particle lithography, which uses the ion beam to exposure resist in a similar fashion to electron beam lithography. Even though FIB-lithography may not be a mainstream FIB technique, a review paper should briefly mention it and point interested readers to relevant papers.

Following reviewer’s suggestion, we have introduced in the text a brief paragraph with the related references (dedicated review and recent applications) to direct the readers interested in the topic.

Lines 119-127

  1. Some chemical formulas (such as (CH3)3Pt(CpCH)3, SiO2, and Si3N4 in the caption of Fig. 1) need correct subscription format.

We double checked the chemical formulas in the revised version.

4.The figure quality of Fig. 3 is poor compared to other figures.

We improved the quality of the figure 3.

5.It is worthwhile to compare the throughput (fabrication speed) of FIB milling and FIB deposition, as this is import for fabricating large area devices and metamaterials.

According to the Referee’s comment we have added a paragraph about the fabrication speed differences in the fabrication of chiral 3Dperiodic nanostructures realized in both milling and induced deposition mode. Lines 403-413

Moreover, it is possible to find other comments in the text in lines 729-736

6.Figure 6 is mentioned before figure 5 in the text. Consider swapping Figs. 5 and 6, or swapping the corresponding sections in the text, or re-arranging the panels in Figs. 5 and 6.

Following the reviewer’s suggestion, we have combined figure 5 and 6 in one image, in order to avoid back and forth values indicated in the related text.

7.The labels of Figs. 6f, 6g, 6h are inconsistent with the description in the caption

Response to answer 3,7. We provided to correct all the typos suggested.

Reviewer 2 Report

Some minor comments on the paper:

  1. It would be an enhancement to the paper to perhaps add a subsection where they discuss a bit more alternative methods of fabricating similar structures. The authors have included a paragraph between lines 834 and 843 but only in the conclusions. This paragraph should perhaps move to an earlier section and expand slightly to better show the pros and cons of FIB compared to the other methods.
  2. Also on line 77 they mention focused electron beam and that this will be treated in another paper. Nevertheless, spending a paragraph on it to compare the two techniques wouldn't be harmful.
  3. There are several typos and missing commas and undefined acronyms (line 160 FEBID, line 469 groove, line 508 bar, line 552 with to, line 680 can movable, line 715 Fig. 8a,,line 780 ORD redefinition, etc)
  4. Some back and forths with Fig. 5 and 6 (Fig. 5 is referenced before Fig. 6)

Author Response

  1. It would be an enhancement to the paper to perhaps add a subsection where they discuss a bit more alternative methods of fabricating similar structures. The authors have included a paragraph between lines 834 and 843 but only in the conclusions. This paragraph should perhaps move to an earlier section and expand slightly to better show the pros and cons of FIB compared to the other methods.

According to the Referee’s comment, we have added a paragraph with the other micro/nano fabrication techniques employed for the realization of 3D chiral photonics nanostructures, a brief focus on their working principle and their advantages and disadvantages.

Lines 862-970

  1. Also on line 77 they mention focused electron beam and that this will be treated in another paper. Nevertheless, spending a paragraph on it to compare the two techniques wouldn't be harmful.

Following the referees suggestion we have added a brief paragraph about focused electron beam. Lines 207-214

Moreover, inside the text it is possible to find some hint about the features of the fabricated 3D chiral nanostructures. Line 364-374, Line 685-688

3. There are several typos and missing commas and undefined acronyms (line 160 FEBID, line 469 groove, line 508 bar, line 552 with to, line 680 can movable, line 715 Fig. 8a,,line 780 ORD redefinition, etc)

4. Some back and forths with Fig. 5 and 6 (Fig. 5 is referenced before Fig. 6)

Questions 3-4. We provided to correct all the typos suggested and the other founded in the text and the incongruities in the figures.